

# All order merging of high energy and soft collinear resummation

Jeppe R. Andersen, Hitham Hassan⋆ and Sebastian Jaskiewicz

Institute for Particle Physics Phenomenology, Durham University, Durham, United Kingdom

⋆ hitham.t.hassan@durham.ac.uk

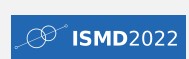

## Abstract

**We present a method of merging the exclusive LO-matched high energy resummation of *High Energy Jets* (HEJ) with the parton shower of PYTHIA which preserves the accuracy of the LO cross sections and the logarithmic accuracy of both resummation schemes across all of phase space. Predictions produced with this merging prescription are presented with comparisons to data from experimental studies and suggestions are made for further observables and experimental cuts which highlight the importance of both high energy and soft-collinear effects.**

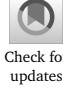

## 1 Introduction

The emergence of complex divergent and slowly-convergent structures in perturbation theory has significant implications on theorists' ability to produce stable and robust predictions for standard model processes. The direct implication of these is that expansions in $\alpha_s$ converge slowly with higher orders of expansion in certain regions of phase space as the size of the logarithms compensates for the smallness of the coupling and an all-order treatment is required. Most pertinent to these proceedings is the presence of large logarithms to all orders of QCD in different kinematic limits.

In the *high energy* (HE) or *multi-Regge kinematic* (MRK) limit for a $2 \rightarrow n$ scattering of partons with momenta $p_a, p_b \rightarrow p_1, p_2 \ldots, p_{n-1}, p_n$:

$$y_1 \gg y_2 \gg \cdots \gg y_{n-1} \gg y_n, \quad p_{i,\perp} \approx k_\perp \quad \forall i \in \{1, 2, \cdots, n-1, n\}, \tag{1}$$

with $k_\perp$ some hard transverse momentum scale, large logarithms $\log(\hat{s}/k_\perp^2) \rightarrow |\Delta y_{1,n}|$ arise to all orders in perturbation theory [1]. Without an accounting for these effects, the phase space corresponding to the MRK limit will be poorly modelled by fixed-order predictions and descriptions of observables which receive this logarithmic enhancement from semi-hard wide-angle radiation will not be perturbatively stable.

These logarithms are most often *resummed* analytically via the Balitsky-Fadin-Kuraev-Lipatov (BFKL) formalism [1–4]. For this study we work instead with the *High Energy Jets* [5] Monte Carlo implementation of high energy resummation which matches the inclusive cross section to the leading order (LO) prediction and applies the *leading logarithmic-* (LL-)accurate high energy corrections to processes which at tree level contribute at LL or NLL accuracy [6].

The HEJ-resummable processes at LL (for pure dijet production) are scatterings of the form $f_1, f_2 \rightarrow f_1, g, \ldots, g, f_2$ where the final state is understood to be ordered in rapidity. We refer to such configurations as "Fadin-Kuraev-Lipatov" or "FKL" configurations. The subleading configurations which receive LL resummation in HEJ include those where the rapidity ordering between any two neighbours in the final state is relaxed.[1] Additional subleading processes include those with $t$-channel quark propagators (allowing for central $q\bar{q}$ emissions). The event output of HEJ is *exclusive* to its logarithmic accuracy compared to the *inclusive* event input generated at leading order.

In the limit of soft-collinear parton splittings a different class of logarithms emerge which typically manifest as ratios of transverse energy scales:

$$\log^2\left(\frac{t_j}{t_k}\right), \tag{2}$$

where $t$ has mass dimension +2. Such double logarithms may be recast into the product of a soft logarithm and a collinear logarithm, each of which diverges respectively when partons in an event have low transverse momenta or have small angular separation. Just as with the high energy logarithms $\log(\hat{s}/k_\perp^2)$, the presence of these spoils the rapid convergence of the perturbative series in the regions of phase space where such logarithms are large. This includes, most plainly expressed in Eq. (2) for a transverse momentum-based evolution scale, configurations with hierarchies in $p_\perp$ and collinear splittings inside jet cones.

These effects may be accounted for with Monte Carlo *parton showers* which resum the leading logarithmic soft-collinear effects to all orders in perturbation theory. Most often *inclusive* fixed-order events are *merged* with parton shower resummation. Merging events generated at fixed-order with parton showers is a well-established field of research with many well-proven procedures including CKKW-L [7, 8] merging for leading order input events. Methods for matching to higher orders in perturbation theory, including MC@NLO [9] and the POWHEG [10] methods for NLO events are also widely used. Further generalisations of these methods for NLO matching which increase the accuracy below the merging scale to full NLO have additionally been developed, including MENLOPS [11, 12] and UNLOPS [13, 14]. The result of these are *exclusive* showered events with an all-order description of the soft and collinear splittings.

We discuss here the implementation of a new procedure for merging the *exclusive* high-energy-resummed event output of HEJ with the *exclusive* parton shower resummation of PYTHIA8 [15] which accounts for both missing higher order perturbative effects and systematically removes the double counted contributions. The software implementation of this procedure is HEJ+PYTHIA.

## 2 Merging Procedure

To produce high energy- and soft-collinear-resummed predictions, we express the resummation of HEJ in the language of the parton shower by defining a splitting kernel corresponding to

---

[1] Sec 1.2.3 of ref [6] provides an overview of contributions from subleading configurations and how these are resummed in HEJ.

the HEJ matrix elements. In regular QCD, the splitting functions may be calculated by: [6,16]:

$$dk_\perp^2 dz \int d\phi \frac{1}{16\pi^2} \frac{\left|\mathcal{M}^{n+1}\right|^2}{|\mathcal{M}^n|^2} \sim \frac{dk_\perp^2}{k_\perp^2} dz \frac{\alpha_s}{2\pi} P(z), \tag{3}$$

with $\mathcal{M}$ the regular matrix elements of QCD. We may calculate the analogue of the splitting function for HEJ analogously, substituting for the regular matrix elements the HEJ-equivalent:

$$P^{\text{HEJ}} = \frac{1}{2} \frac{1}{16\pi^2} \frac{\overline{\left|\mathcal{M}_{\text{HEJ}}^{n+1}\right|^2}}{\left|\mathcal{M}_{\text{HEJ}}^n\right|^2}, \tag{4}$$

where the extra factor of $1/2$ in the HEJ splitting probability arises when we consider colour configurations. There are two possible colour configurations for inserting a particle into a HE configuration [17,18] which we weight equally in our treatment. The quantities highlighted in blue from Eq. (3) are implicitly contained in $P^{\text{HEJ}}$. Our method borrows from CKKWL merging in a similar manner to ref. [19] by introducing a scheme within which shower emissions are vetoed according to the probability they have already been accounted for by HEJ:

$$\mathcal{P}^{\text{veto}} = \frac{P^{\text{HEJ}}}{P^{\text{PYTHIA}}} \cdot \Theta\left(P^{\text{PYTHIA}} - P^{\text{HEJ}}\right) + 1 \cdot \Theta\left(-P^{\text{PYTHIA}} + P^{\text{HEJ}}\right), \tag{5}$$

with $P^{\text{PYTHIA}}$ the Altarelli-Parisi splitting kernels of QCD (weighted by $\alpha_s/2\pi k_\perp^2$). For the case of an initial state emission $i \to jk$, the splitting kernel must be reweighted by the correct ratio of parton distribution functions (PDFs) to properly reproduce the backwards DGLAP evolution:

$$P \to P \cdot \frac{x_i f_i(x_i, \mu_F^2)}{x_j f_j(x_j, \mu_F^2)}, \tag{6}$$

with $f_{i,j}$ the relevant PDF at energy fraction $x_{i,j}$ and an appropriately chosen factorisation scale $\mu_F$. The combined effect of the above is that for emissions with $P^{\text{PYTHIA}} < P^{\text{HEJ}}$ we veto trial emissions with 100% probability and thus the shower emissions are produced with a modified Sudakov form factor:

$$\Delta^S(k_{\perp,i}^2, k_{\perp,i+1}^2) = \exp\left\{-\int_{k_{\perp,i+1}^2}^{k_{\perp,i}^2} dk_\perp^2 dz \, \Theta\left(P^{\text{PYTHIA}} - P^{\text{HEJ}}\right)\left[P^{\text{PYTHIA}}(k_\perp^2, z) - P^{\text{HEJ}}(k_\perp^2, z)\right]\right\}, \tag{7}$$

which subtracts from the PYTHIA splitting kernel the equivalent for HEJ.

For HEJ-resummable input events, we construct *histories* - sequences of splittings that connect the LO $2 \to 2$ scattering (for the case of inclusive dijet production) to the $2 \to n$ input event, ordered in the PYTHIA shower evolution variable $k_\perp^2$. We assign to each history a weight proportional to the product of HEJ splitting probabilities and select one according to their weights. Our procedure for merging such configurations may be summarised as follows:

1. Start the trial shower with an emission from state $i$ in the history with scale $k_{\perp,i}^2$ (starting with $i = 0$).

    (a) if $k_{\perp,i+1}^2 < k_{\perp,i}^2$, continue to the next state in the history, setting $i \to i+1$ and returning to step 1. If this is the original event, go to step 2.

    (b) if $k_{\perp,i+1}^2 > k_{\perp,i}^2$, veto the emission with probability $\mathcal{P}^{\text{veto}}$ of Eq. (5).

        i. If vetoed generate a new trial emission from the current scale and return to step 1a.

      ii. If not vetoed, **keep the trial emission** and append it to each subsequent node in the history $i + 1, i + 2, \ldots$ . Generate a new trial emission from the current scale and return to step 1a.

2. Perform a final trial emission (to account for the case no emissions have yet been appended due to the veto) and exit the merging after accepting an emission according to the veto probability in 1b, using the updated event record to initiate the later shower.

3. Continue the parton shower evolution on the merged event (i.e. the input HEJ event dressed with the additional shower emissions at all stages of the history), veto each trial emission with probability $\mathcal{P}^{\mathrm{veto}}$ until the hadronisation scale is reached.

4. Hadronise the event.

We note several developments since the previous merging procedure of ref. [19] (which itself was a development of ref. [20]). Most noticeably, all of the original HEJ partons from the input event are retained (up to later splittings) in our merging procedure meaning that the coverage of phase space probed during high energy resummation is conserved. In the previous implementation the shower phase space was unrestricted, but the merging procedure would terminate after accepting one emission from PYTHIA thus not guaranteeing the retention of high energy or shower logarithmic accuracy beyond the first perturbative correction.

    Secondly, much development in the HEJ formalism has allowed for a greater number of configurations to be resummed including those contributing to subleading logarithmic accuracies in BFKL [6]. This means the definition of HEJ-states as they are referred to in ref. [19] has been widened to include configurations with a central/extremal (in rapidity) $q\bar{q}$ pair or an extremal gluon at LO as we have discussed.

    Thirdly, the non-HEJ-resummable states at LO are merged via ordinary CKKWL in PYTHIA as this procedure has been designed to account for the double counting of emissions. We thus retain the inclusive cross section and implement a completely unitary procedure which perserves the logarithmic accuracy of both resummation schemes to all orders in perturbation theory across phase space.

## 3 Results

We present here validation predictions for the HEJ+PYTHIA merging procedure by comparing our results to experimental data from the ATLAS collaboration of jet profiles for inclusive jet production at $\sqrt{s} = 7$ TeV [21]. Jets were clustered with a radius parameter $R = 0.6$ and a minimum transverse momentum of $p_{\perp,j} > 30$ GeV, with the additional requirement that they were central in rapidity i.e. $|y_j| < 2.8$. The differential jet profile $\rho(r)$ is defined by:

$$\rho(r) = \frac{1}{\Delta r} \frac{1}{N_{\mathrm{jets}}} \sum_{\mathrm{jets}} \frac{p_T(r - \Delta r/2, r + \Delta r/2)}{p_T(0, R)}, \tag{8}$$

with $R$ the jet radius parameter and $p_T(r_1, r_2)$ the (scalar) sum of the transverse momentum in an annulus (in $y$-$\phi$ space) between radii $r_1$ and $r_2$. The integrated jet profile $\Psi(r)$ is defined as the normalised integral of $\rho$ between 0 and $r$.

    The region of phase space probed by such measurements receives strong soft and collinear enhancement and small corrections from BFKL effects. Indeed, HEJ predicts jet profiles overwhelmingly dominated by one central hard parton as demonstrated in Fig. 1. Parton showers excel in their description of such observables as they resum the very effects required to populate the regions of phase space probed - thus we expect little difference between a pure PYTHIA prediction and our merged HEJ+PYTHIA prediction. This is indeed the trend shown in Fig. 1

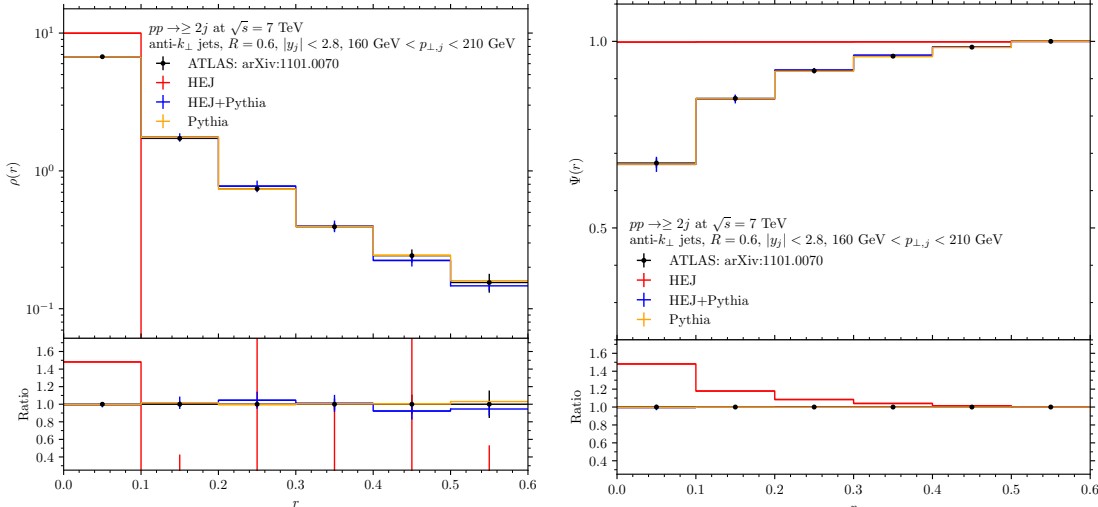

Figure 1: Comparisons of HEJ, HEJ+PYTHIA and PYTHIA to experimental data for differential jet profiles (left) and integrated jet profiles (right) with a hard jet selection. The showered predictions have MPI and hadronisation enabled with "Monash 2013" the chosen tune. Data from ATLAS [21] - the cuts of this experimental analysis are displayed on the figure.

with both showered predictions yielding robust descriptions of the ATLAS data [21]. This is a reassuring validation test of our algorithm and demonstrates that we are able to dress HEJ-resummed states with the missing higher order shower logarithms.

## 4 Conclusion

The first validation tests of our merging procedure which remains in development are highly encouraging. This study has proved further that the intersection of different resummation schemes is rarely straightforward and treating the two regimes discussed in this study as "opposites" does not lend itself to producing physical descriptions of standard model processes [19].

Our aims for this work are to extend the formalism to include $H$, $Z/\gamma$, and $W$ production with jets such that all processes treated by HEJ may be produced with physical shower evolution — bringing the hard partonic predictions of HEJ closer to experimental truth-level. We additionally seek to produce predictions for distributed cross sections and ratios (e.g. $R_{32}$) in terms of the pertinent experimental observables to these resummation schemes. These include (and are not limited to) $p_\perp$-based observables for the shower predictions and dijet invariant masses and rapidity differences to test the preservation of the HEJ logarithmic accuracy.

The types of experimental analysis which would probe regions of phase space with high energy and soft-collinear enhancement would include an inclusive set of cuts such as the following:

1. Wide rapidity selection e.g. $|y| < 4.5$ rather than restricting to the central region $|y| < 2.8$ — this will test regions where HEJ corrections are applied.

2. Hard jets with $p_\perp$-hierarchy in jet selection e.g. $p_{\perp,j} > 60$ GeV, with $p_{\perp,j_1} > 80, 100, 120$ GeV — this will test the correct application of shower corrections, varying this gives an indication of the size of the effect.

3. Jets of varying radii ($R = 0.4, 0.6$) — this will explore the effect of broadening the jet cone on each resummation scheme.

This will demonstrate that neither resummation on its own will stably model the entirety of the available phase space and explore the method we have developed for properly accounting for both missing higher order effects.

## Acknowledgements

We would like to thank the organising committe for inviting us to present our research and for organising a most productive and engaging conference. We also thank our colleagues in the HEJ collaboration for our discussions during this project.

**Funding information**   HH gratefully acknowledges funding by the UK STFC under grant ST/T506047/1.

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
