# Peer review of "All Order Merging of High Energy and Soft Collinear Resummation"

_SciPost Physics Proceedings, doi:SciPost Phys. Proc. 15, 022 (2024)_

## Round 1 · Referee Report · Anonymous (Referee 1) · 2023-1-11

Report

The author reports on a new method for merging the exclusive LO-matched high energy resummation of High Energy Jets (HEJ) with the PYTHIA parton shower that preserves the logarithmic accuracy of both resummation schemes. This work constitutes an improvement over the approach developed by one of the authors in Ref [19]. The paper is well written and constitute a scientifically sound and original work. I recommend its publication.

I will have a few minor comments and questions though that may help clarify the text.
1- Define k_T in Eq. (1). 2- Define "FKL" 3- In the 1st paragraph page 2, " The subleading configurations which receive LL resummation in HEJ ..." do the author mean NLL resummation since these configurations are sub-leading ? 4- Eq. (2) is a bit confusing: I don't see how the double log structure on the r.h.s. emerges from the l.h.s. that displays a single log? 5- To me Eq (3) corresponds to the splitting probability in PYTHIA so it's not clear to me at all what the difference between P^{HEJ} and P^{PYTHIA} is. A clarification on this would be useful.

  • validity: high
  • significance: high
  • originality: high
  • clarity: high
  • formatting: excellent
  • grammar: excellent

Author:  Hitham Hassan  on 2023-01-20  [id 3251]

(in reply to Report 1 on 2023-01-11)

Dear Referee,

Many thanks for your comments and detailed feedback, we have updated our draft and aimed to address the points raised in your report. We summarise our responses to each point additionally in this message:

  1. k_T in Eq.(1) is a transverse momentum scale which all jets in the event are of a similar hardness to in the MRK limit, we have added a definition to the proceedings.
  2. "FKL" refers to "Fadin-Kuraev-Lipatov", three of the founders of the BFKL formalism, and we use this naming concention to identify the LL configurations in the MRK limit. We have added a note to this effect to the proceedings.
  3. "The subleading configurations which receive LL resummation in HEJ ...": The configurations in question contribute to the NLL corrections to dijet observables. In the BFKL language, the full set of NLL corrections consists of real and virtual corrections to both impact factors and the evolution. The contributions in question relate to the real emission contribution (two-jet configuration) to the impact factors and the evolution, i.e. start contributing at 3 and 4 jet born configuration. Leading logarithmic resummation is then applied to these already sub-leading configurations.
  4. The soft-collinear logarithm: In the previous version of the draft we had incorrectly omitted a power of two from the logarithm on the LHS. In response, we have reformulated the equation in more general terms, referring to a generic evolution scale $t$ rather than a specific transverse momentum. Additionally, we have written a explanation that the double logarithmic structure of the ratio of evolution scales may be recast into a soft and collinear logarithm; each individually divergent.
  5. Confusion around Eq.(3): The expression in Eq.(3) in the original draft is confusing and we have followed the recommendation to alter it. We replace Eq.(3) with a general expression for splitting functions in QCD (which are used in PYTHIA) and have added an equation (and some description) defining the HEJ analogue, making use of the HEJ-resummed matrix elements.

Thank you again for your review and report, the comments have been very useful and facilitated productive discussion between us. Please let us know if there is anything else we can clarify or if we can address the points above differently.

Best regards, Hitham, Jeppe, and Sebastian

---

## Round 2 · Author Response

We thank the reviewer for their insightful comments and have updated the proceedings to incorporate the changes suggested. The full list of changes is also provided.

---

## Round 2 · List of Changes

k_T in Eq.(1): This is a transverse momentum scale which all jets in the event are of a similar hardness to in the MRK limit, we have added a definition to the proceedings.

"FKL" refers to "Fadin-Kuraev-Lipatov", three of the founders of the BFKL formalism, and we use this naming convention to identify the LL configurations in the MRK limit. We have added a note to this effect to the proceedings.

The soft-collinear logarithm, Eq.(2): In the previous version of the draft we had incorrectly omitted a power of two from the logarithm on the LHS. In response, we have reformulated the equation in more general terms, referring to a generic evolution scale $t$ rather than a specific transverse momentum. Additionally, we have written a explanation that the double logarithmic structure of the ratio of evolution scales may be recast into a soft and collinear logarithm; each individually divergent.

Confusion around Eq.(3): The expression in Eq.(3) in the original draft is confusing and we have followed the recommendation to alter it. We replace Eq.(3) with a general expression for splitting functions in QCD (which are used in PYTHIA) and have added an equation (and some description) defining the HEJ analogue, making use of the HEJ-resummed matrix elements.

---

## Editorial Decision

published